

# Rare patterns of dorsal puncture in *Pterostichus oblongopunctatus* (Coleoptera: Carabidae)

Axel Schwerk[1] and Radomir Jaskuła[2]

[1] Laboratory of Evaluation and Assessment of Natural Resources, Faculty of Horticulture, Biotechnology and Landscape Architecture, Warsaw University of Life Sciences—SGGW, Warsaw, Poland
[2] Department of Invertebrate Zoology and Hydrobiology, Faculty of Biology and Environmental Protection, University of Lodz, Łódź, Poland

## ABSTRACT

**Background**. The carabid beetle species *Pterostichus oblongopunctatus* is common in different types of forests in Poland and Europe. With respect to this species, some unclarities exist concerning the morphological feature of punctures on the elytra. *P. oblongopunctatus* has dorsal pits in the third interval of the elytra, the available identification keys, however, provide inconsistent information concerning the puncture in other intervals. During long-term studies at different study sites in Poland, the first author rarely but regularly discovered individuals with unusual dorsal puncture patterns, i.e., pits in the fifth and even in the seventh interval of the elytra. Since such rare patterns might be connected with special habitat characteristics, and thus have a potential as an indicator, the aim of the study was to test if they are connected with specific subpopulations (interaction groups), if they are related to the sex or size of the beetles, and if they are related to specific habitat conditions.

**Material and Methods**. We counted the pits on the elytra, determined the sex, and measured the length of the right elytron of individuals of *P. oblongopunctatus* collected at numerous study sites located within the borders of the Regional Directory of National Forests in Piła (Western Poland) over the period 2014–2016.

**Results**. Altogether, 1,058 individuals of *P. oblongopunctatus* were subjected to statistical analysis. Almost 19% of the individuals had a dorsal puncture in the fifth interval of the elytra and about 0.7% had a dorsal puncture in the seventh interval of the elytra. In 2014 and 2015, significantly more females exhibited such unusual patterns of dorsal puncture than males. Even if not statistically significant, in 2016 also relatively more females showed such a pattern. Neither males nor females of the analysed individuals with usual puncture patterns showed a significant difference in the length of the right elytron from those with unusual puncture patterns, and neither for males nor for females a significant correlation of the percentage share of the individuals with unusual puncture patterns with the age of the study sites could be detected. However, both males and females with unusual patterns had more dorsal pits than those without. Moreover, males as well as females showed in all those years a trend that the individuals with unusual patterns have more pits in the third interval of the elytra.

**Discussion**. The results indicate that females are more likely to exhibit unusual patterns. Since individuals of *P. oblongopunctatus* with a higher number of pits on the elytra are supposed to prevail in more wet habitats, such patterns might be related to moisture conditions. The possibility of pits in the seventh interval of the elytra should be added to identification keys.

Corresponding author
Axel Schwerk, aschwerk@yahoo.de

## INTRODUCTION

The carabid beetle species *Pterostichus oblongopunctatus* (Fabricius, 1787) is common in different types of forests in Poland and Europe. Several studies deal with this species. Particularly Szyszko (e.g.: *Szyszko, 1976*; *Szyszko, Vermeulen & Schäffer, 1992*; *Vermeulen & Szyszko, 1992*; *Szyszko, Vermeulen & Den Boer, 1996*) has studied it intensively. Generally, it could be demonstrated that successional changes in its habitats are accompanied by modifications in the life-history pattern of this species (*Szyszko, Vermeulen & Den Boer, 1996*). Other studies dealt, among others, with aspects of regulation (*Brunsting & Heessen, 1984*), habitat selection (*Brunsting, 1981*; *Paje & Mossakowski, 1984*; *Brygadyrenko, 2016*), or exposure to environmental stressors such as metals or pesticides (*Bednarska & Laskowski, 2009*; *Szyszko, Schwerk & Płatek, 2010*; *Bednarska, Brzeska & Laskowski, 2011*; *Bednarska & Stachowiz, 2013*; *Skalski et al., 2015*).

Although the ecological demands of *P. oblongopunctatus* are well known, some unclarities exist concerning the morphological feature of punctures on the elytra. The species has regular and well-pronounced pits in the third interval of the elytra, but with respect to the puncture in other intervals of the elytra, identification keys provide inconsistent information. *Reitter (1908)*, *Mrozek-Dahl (1928)*, *Lindroth (1986)* and *Müller-Motzfeld (2004)* specify pits only in the third interval of the elytra. *Trautner & Geigenmüller (1987)*, however, mention "usually only the 3rd elytral interval with dorsal punctures, rarely the 5th too", and *Hůrka (1996)* specifies "2–9 in interval 3, 0–2 in interval 5". None of these keys mentions the possibility of pits in the seventh interval.

Concerning the puncture patterns of this species, *Den Boer, Szyszko & Vermeulen (1993)* distinguish the so-called "high pitters" with many pits on the elytra and the so-called "low pitters" with a low number of pits. They have demonstrated that the number pits is related to the moisture conditions in the habitat during the period of larval development, with "high pitters" being characteristic of more moist sites. Rearing experiments indicated that low- and high-pitters corresponded with two groups of genotypes which are fully expressed in the two patterns (*Den Boer, Szyszko & Vermeulen, 1993*). *Emetz (1984)* explained shifts of the phenotypical composition and the level of asymmetry of dorsal pits in a *P. oblongopunctatus* population in an oak forest in the suburbs of Voronezh with response to changes of environmental conditions due to recreation pressure. He argues that increased recreation pressure may have impact on the litter layer and cause deterioration of the soil water regime.

During long-term studies at different study sites in Poland, the first author rarely but regularly discovered individuals with an unusual dorsal puncture, i.e., pits in the fifth and even in the seventh interval of the elytra (Fig. 1). Such rare patterns might be connected with special habitat characteristics, and thus have a potential as an indicator. Therefore, the aim of the presented paper was to analyse the occurrence of pits in the fifth and seventh

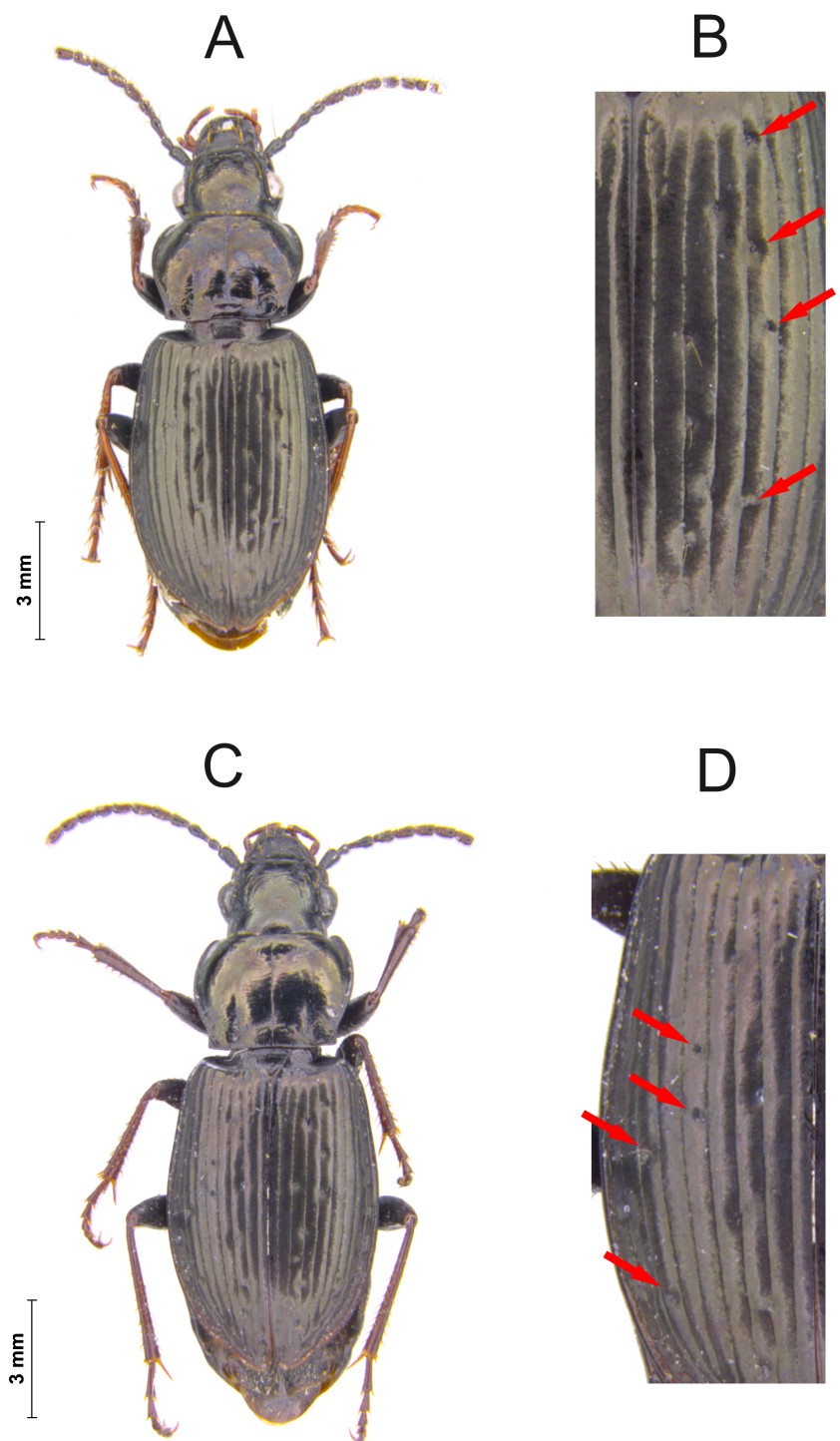

**Figure 1** **Individuals of *P. oblongopunctatus* with unusual dorsal puncture.** Female with four pits in the fifth interval (A—full individual, B—detail) and female with pits in the fifth and seventh intervall (C—full individual, D—detail). Arrows indicate pits (Photo credit: R. Jaskuła).

interval of the elytra of *P. oblongopunctatus*. In particular, we wanted to test the following hypotheses:

(1) The unusual puncture patterns are connected with specific subpopulations (interaction groups) and will occur repeatedly at the same study sites over the years.

(2) The unusual patterns are related to the sex of the beetles. Pits in the fifth and seventh interval will occur in one sex only or significantly more often in one sex.

(3) The unusual patterns are related to the size of the beetles, i.e., individuals with pits in the fifth and seventh interval should be bigger than those without.

(4) The unusual patterns are related to the stage of succession of the study sites and are, therefore, correlated with the age of the sites.

(5) The unusual puncture patterns are related to the moisture conditions of the habitat and exist more often in high pitters. Individuals with unusual puncture patterns should have higher numbers of pits in general as well as in the third interval of the elytra.

## MATERIAL AND METHODS

### Field methods and data elaboration

The material analysed was collected using pitfall traps in the framework of a long-term study in six research areas in the Regional Directorate of the State Forests in Piła, Western Poland: research area ''Martew'' (68 study sites), research area ''Potrzebowice'' (41 study sites), research area ''Krzywda'' (26 study sites), research area ''Trzcianka'' (three study sites), research area ''Kaczory'' (two study sites), and Research area ''Zdrojowa Góra'' (two study sites). Detailed information about the research areas and study sites is provided by *Schwerk (2008)* and *Dymitryszyn, Szyszko & Rylke (2013)*. The majority of the study sites are forest stands of different age. However, nine of the study sites in the research area ''Krzywda'' are located on fallow land.

Pitfall traps were glass jars, which were installed from mid-May to mid-September, with pure ethylene glycol as trapping liquid. A funnel with a diameter of ca. 10 cm was installed over each trap flush with the soil surface to minimise by-catch, and a roof was installed a few centimetres above the funnel to protect the trap from rainfall.

Detailed studies were carried out on individuals collected in the years 2014–2016. The sex of the specimens was determined and the puncture patterns of all the individuals of *P. oblongopunctatus* were registered by counting the pits in the third, fifth and seventh interval on both elytra using a stereo microscope. Additionally, the length of the right elytron of each individual was measured as an indicator of body size. The studied individuals were levelled out under a stereo microscope at $14\times$ magnification and the length was measured using a slide calliper with an accuracy of 0.01 mm.

Digital photographs of the characteristic individuals were taken using a digital camera Leica DFC295 connected with a Leica M205C stereomicroscope and compiled by LAS v.4.5 Ink. software.

The studied material was elaborated within the scope of the project no OR-2717-21/14, General Directorate of the State Forests.

## Statistical methods

In order to study to which degree unusual puncture patterns were repeated at specific study sites, the number of sites at which in at least one year of the study such patterns were detected was counted. Next the percentage share of study sites with the occurrence of unusual patterns in at least two years and the study sites with the occurrence of unusual patterns in all three years was calculated.

The distribution of individuals with unusual puncture patterns representing both sexes was tested using Chi-square-tests (*Sachs, 1984*). This was done for all the years of study separately as well as for the data of all the years pooled.

Preliminary statistical analyses (Mann–Whitney $U$-tests) showed that females were generally significantly bigger than males in all the analysed years. Moreover, females had significantly more pits than males in all those years, and females had significantly more pits in the third interval than males in 2014 and 2016, and a clear trend concerning this matter in 2015. Therefore, statistical analyses were carried out for males and females separately. All analyses were carried out for the individual years of the study separately as well as for the data of all the years pooled, with the exception of studying the relation to age classes of the study sites, which was done for the pooled data only due to a low number of collected specimens in the youngest and oldest age class.

The size distribution of the individuals without unusual puncture patterns was compared with those with unusual puncture patterns using the Mann–Whitney $U$-test (*Sachs, 1984*).

The relationship of the unusual individuals with the age of the study sites was studied for the research area "Martew", which has forest stands of sufficiently diversified age structure. Based on their age in the respective year of the study, the study sites were assigned to the following age classes according to the system of the Polish State Forests (*Lasy Państwowe, 2012*): 1–20 years, 21–40 years, 41–60 years, 61–80 years, and 81–100 years. The percentage share of the individuals with unusual puncture patterns in the total number of specimens collected was calculated for each age class for males and females separately. Correlations of the percentage share of the unusual individuals with the age class were tested using the Spearman rang correlation coefficient.

The total pit numbers as well as the numbers of pits in the third interval of the elytra of the individuals without unusual puncture patterns were compared with the respective values for the individuals with unusual puncture patterns using the Mann–Whitney $U$-test (*Sachs, 1984*).

Statistical analyses were carried out using IBM SPSS Statistics, version 23.

## RESULTS

Altogether, 1,071 specimens (346 individuals in 2014, 417 in 2015, and 308 in 2016) of *P. oblongopunctatus* were collected. Of those, 13 specimens were excluded from all statistical analyses due to missing or damaged elytra. Almost 19% of the remaining 1,058 beetles had a dorsal puncture in the fifth interval and about 0.7% of the analysed individuals had dorsal pits in the seventh interval of the elytra. Since for seven of the 1,058 specimens the sex could not be determined and one individual had a damaged right elytron, 1,051 beetles

**Table 1 Numbers of individuals of *P. oblongopunctatus* included in statistical analyses.** Question marks indicate individuals, for which sex could not be determined and which were excluded from analysis taking into account sex. Numbers of brackets are due to an individual, which was excluded from analyses taking into account elytra length because of a damaged elytra.

| Type of individuals | 2014 | | | 2015 | | | 2016 | | | All years | | |
|---|---|---|---|---|---|---|---|---|---|---|---|---|
| | ♂ | ♀ | ? | ♂ | ♀ | ? | ♂ | ♀ | ? | ♂ | ♀ | ? |
| Usual | 97 | 182 (181) | 3 | 128 | 206 | – | 99 | 139 | 2 | 324 | 527 (526) | 5 |
| Unusual (pits in interval 5) | 7 | 51 | – | 15 | 61 | – | 19 | 40 | 2 | 41 | 152 | 2 |
| Unusual (pits in interval 5 and 7) | – | 1 | – | – | 2 | – | – | 1 | – | – | 4 | – |
| Unusual (pits in interval 7) | 1 | 1 | – | – | 1 | – | – | 1 | – | 1 | 2 | – |
| Sum | 105 | 235 (234) | 3 | 143 | 270 | – | 118 | 180 | 4 | 366 | 685 (684) | 7 |

**Table 2 Numbers of study sites and numbers of *P. oblongopunctatus* for the individual research areas.** Number of study sites (A), number of study sites with *P. oblongopunctatus* collected (B), number of study sites with individuals of *P. oblongopunctatus* with unusual puncture pattern in at least one year (C), number of study sites with individuals of *P. oblongopunctatus* with unusual puncture pattern in at least two years (D), and number of study sites with individuals of *P. oblongopunctatus* with unusual puncture pattern in all three years of the study (E).

| Research area | A | B | C | D | E |
|---|---|---|---|---|---|
| "Martew" | 68 | 65 | 45 | 20 | 3 |
| "Potrzebowice" | 41 | 38 | 20 | 4 | 1 |
| "Krzywda" | 26 | 19 | 11 | 6 | 3 |
| "Trzcianka" | 3 | 1 | 1 | 1 | 0 |
| "Kaczory" | 2 | 2 | 2 | 1 | 0 |
| "Zdrojowa Góra" | 2 | 2 | 2 | 2 | 1 |
| Sum | 142 | 127 | 81 | 33 | 8 |

were subjected to statistical analyses involving the sex and 1,050 individuals were taken into account when considering both sex and size (Table 1).

*P. oblongopunctatus* was collected at least in one year of the study at 127 of the 142 study sites. At 81 of those study sites at least in one year an unusual individual was collected (Table 2). At 33 (40.7%) of those study sites where at least in one year an unusual individual was collected such specimens were collected in at least 2 years. In all the three years at eight (9.9%) of the study sites where at least in one year an unusual individual was collected such specimens were collected. However, the results differed between the research areas (cf. Table 2).

The percentage of unusual puncture patterns for males was 7.6% (2014), 10.5% (2015), and 16.1% (2016). The respective values for females were 22.6% (2014), 23.7% (2015), and 22.8% (2016). Taking into account all the years together, 11.5% of males and 23.1% of females showed unusual dorsal puncture patterns. Differences between males and females were significant for 2014, 2015 and for the pooled data of all the years of study (Figs. 2A–2B).

In the study of all the individual years as well as data for all the years pooled, both males and females of specimens with usual puncture patterns did not show a significant difference in the length of the right elytron from those with unusual puncture patterns (Fig. 3).

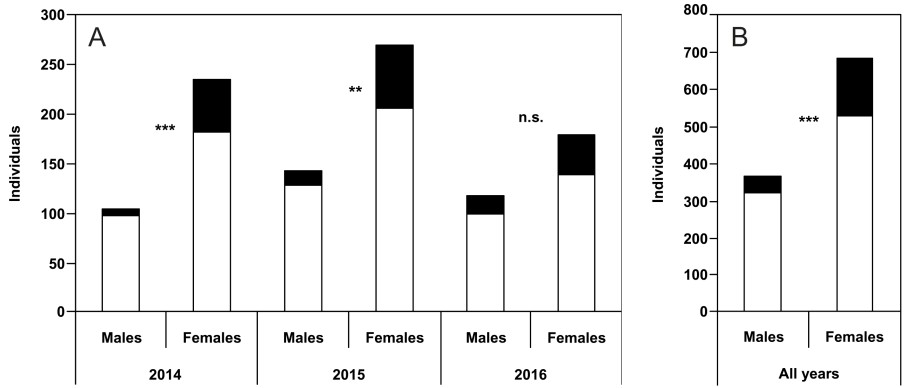

**Figure 2** **Numbers of males and females of *P. oblongopunctatus* in the years of study and all years together with the share of individuals with unusual puncture patterns indicated in black.** Differences between males and females—Chi-Square tests: ***, $p < 0.001$; **, $p < 0.01$; n.s., not significant.

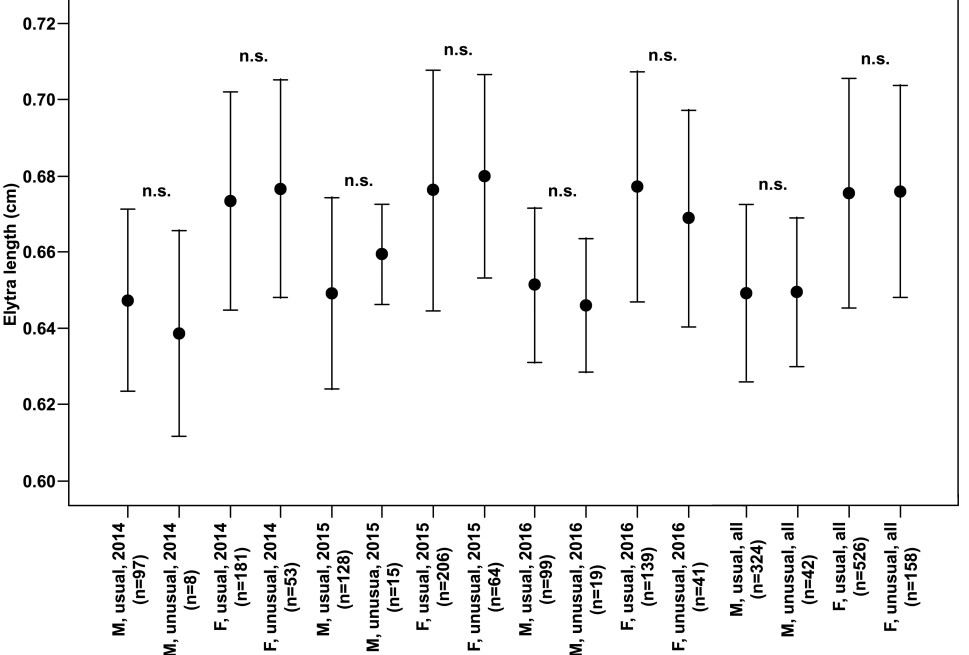

**Figure 3** **Elytra length (Mean ± SD) of males and females of usual and unusual individuals of *P. oblongopunctatus* in the years of study and all years together.** Differences between usual and unusual individuals—Mann–Whitney *U* tests: n.s., not significant.

Males showed a continuous increase in the percentage share of the individuals with unusual puncture patterns from the youngest age class to forest of 61–80 years, but for the oldest age class (81–100 years) a strong decline was visible (Fig. 4A). Females showed an increase in the percentage share of the individuals with unusual puncture patterns from the youngest forests to those of 41–60 years of age, but in the age class of 61–80 years

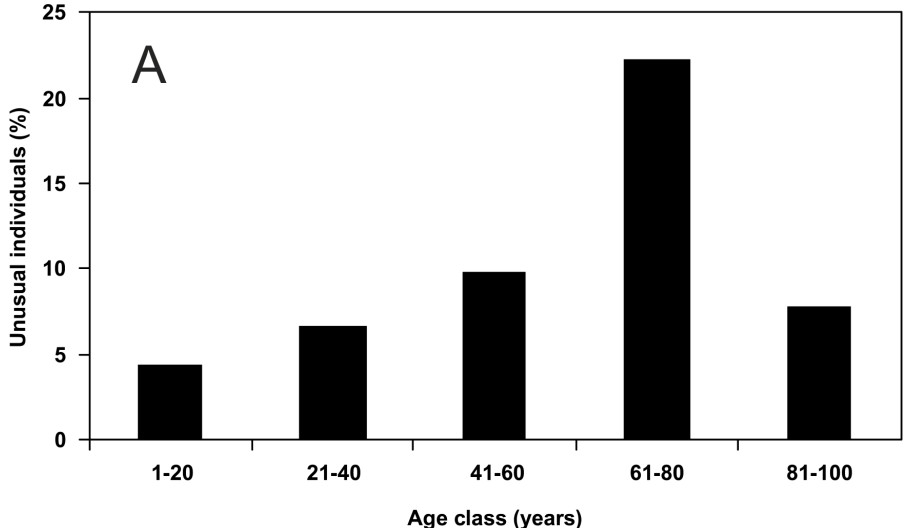

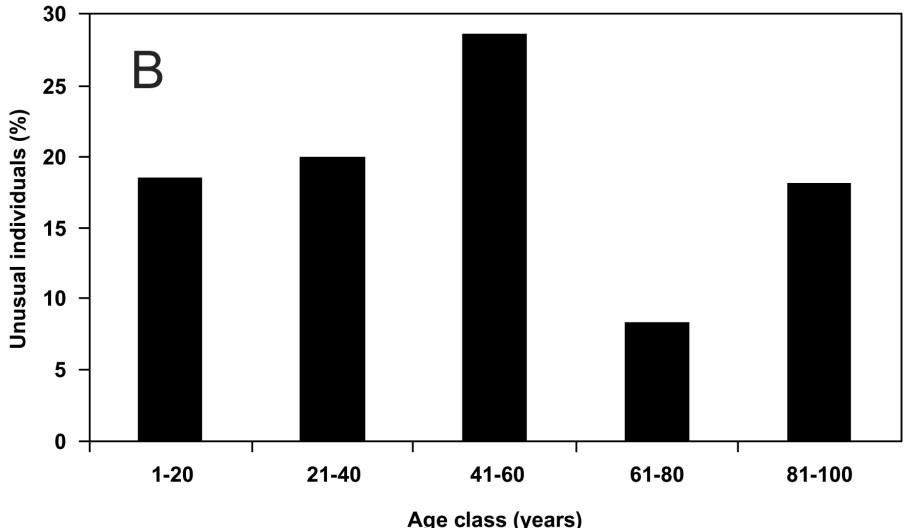

**Figure 4** **Percentage share of unusual individuals of *P. oblongopunctatus* on the total of individuals collected in forest stands of different age classes in the forest range Martew.** (A) Males (Spearman rang correlation coefficient: $r = 0.700$, not significant), (B) females (Spearman rang correlation coefficient: $r = -0.500$, not significant).

the percentage share dropped significantly. Forest of 81–100 years showed an increase compared to the previous age class (Fig. 4B). Neither for males nor for females a significant correlation of the percentage share of the individuals with unusual puncture patterns with the age of the study sites could be detected.

Both males and females of specimens with unusual puncture patterns had significantly more pits on the elytra than those with usual puncture patterns in each year of the study separately and for data of all the years pooled (Fig. 5). When taking into account only

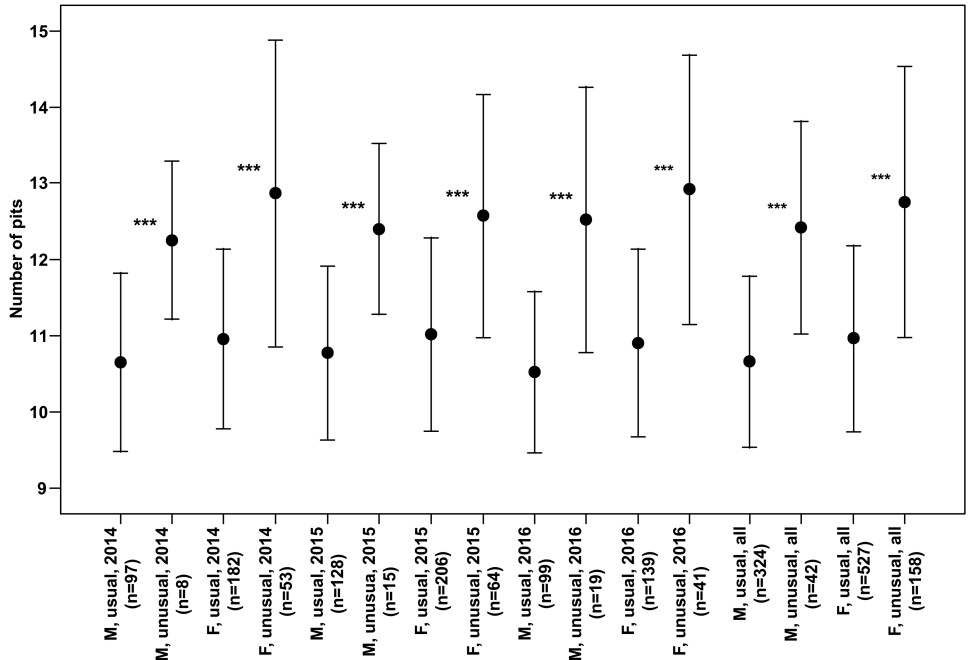

**Figure 5** **Numbers of pits (Mean ± SD) of males and females of usual and unusual individuals of *P. oblongopunctatus* in the years of study and all years together.** Differences between usual and unusual individuals—Mann–Whitney $U$ tests: ***, $p < 0.001$.

pits in the third interval, no significant results could be detected for the individual years of study. However, both males and females with unusual patterns had regularly higher numbers with a clear trend ($p < 0.1$) for males in 2014 and females in 2016. Taking into account the data of all the years pooled, both males and females of the individuals with unusual patterns had significantly higher numbers of pits in the third interval (Fig. 6).

## DISCUSSION

The results of the study indicate that unusual puncture patterns seem not to be related to special subpopulations (interaction groups) of *Pterostichus oblongopunctatus* (hyp. 1) and are not correlated with the size of the analysed individuals (hyp. 3) or the age of the study sites (hyp. 4) in males or females. However, unusual puncture patterns existed significantly more often in females (hyp. 2), which were also significantly bigger than males, and the individuals with unusual puncture patterns had significantly more pits on the total elytra and in the third interval of the elytra (hyp. 5).

The rather low share of study sites which showed repeatedly unusual puncture patterns in two or three years of the study indicates that these patterns are not related to certain sites. Thus, if the patterns are related to a possible environmental factor, this factor may fluctuate at the study sites and the appearance of unusual patterns expresses these fluctuations. Adaption to such stochastic fluctuations in environmental factors was described by *Den Boer (1968)* as "spreading of risk".

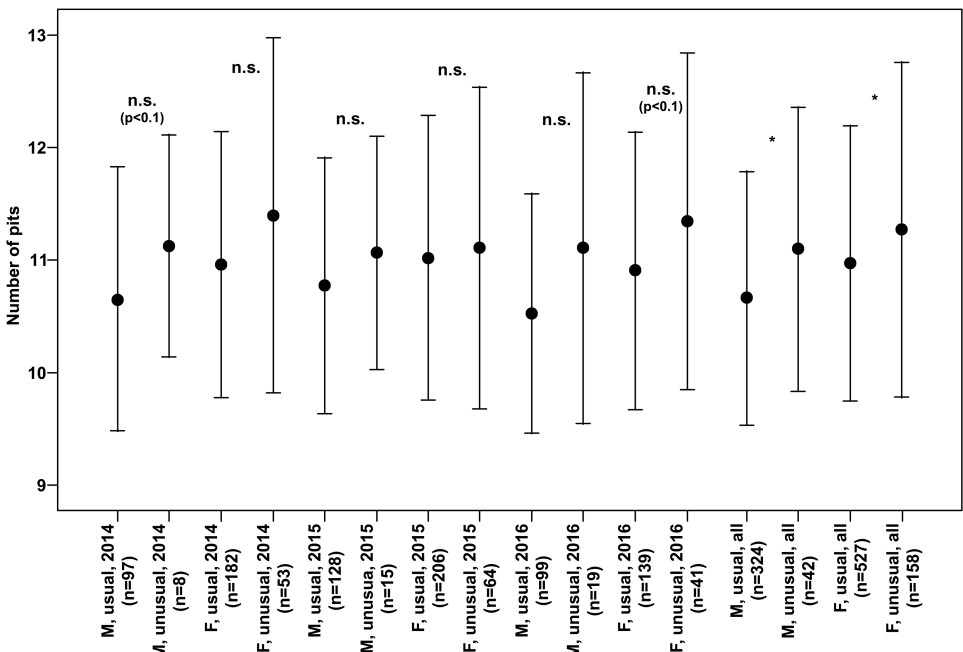

**Figure 6** Numbers of pits in the third interval of the elytra (Mean ± SD) of males and females of usual and unusual individuals of *P. oblongopunctatus* in the years of study and all years together. Differences between usual and unusual individuals—Mann–Whitney *U* tests: *, $p < 0.05$; n.s., not significant.

However, unusual puncture patterns were observed more frequently in females than in males, which may be explained by the fact that females are significantly bigger than males. Accordingly, females had also more pits both in all the intervals of the elytra and in the third interval only. Yet, a dependency of unusual puncture patterns on the size should be also visible in males and females independently, and both in males and females no such relation could be observed. *Szyszko, Vermeulen & Den Boer (1996)* showed that individuals in advanced stages of succession of the habitat were significantly bigger. Therefore, the missing correlation of unusual puncture patterns with the size of the individuals in both sexes indicates that such patterns are not related to the stage of succession. This conclusion is supported by the result showing that no correlation of increasing or decreasing share of the individuals with unusual patterns with the age of the study sites could be observed.

The result indicating that individuals with unusual puncture patterns had more pits in general, even in the third interval of the elytra, supports the hypothesis that those patterns are related to the moisture conditions during the period of larval development. Like high pitters (*Den Boer, Szyszko & Vermeulen, 1993*), unusual puncture patterns seem to indicate moister habitat conditions. In conclusion, a high percentage share of the individuals with unusual patterns may be used as an indicator of moisture conditions. In *P. oblongopunctatus* hygroreceptor neurons are known for their antennal dome-shaped sensilla (*Merivee et al., 2010*). Although currently there is a lack of clear evidence that the setae in elytral pits in *P. oblongopunctatus* play a role of hygroreceptors too, single data from some other insect groups (in which setae are present on the abdomen and/or on ovipositor) suggest that

it also cannot be excluded (*Bell & Cardé, 1985*). A higher number of setae on the elytra due to a higher number of pits in the fifth and seventh intervals in females compared to males, as shown in our study, may be explained by the role of this sex in the reproductive behaviour. As in all insect species, also females of *P. oblongopunctatus* need to find a proper place to deposit their eggs after copulation. This is one of the most crucial conditions in females' post-copulatory reproductive behaviour as it determines developmental success of their embryos (*Thornhill & Alcock, 1983*). In the case of *P. oblongopunctatus*, a typically epigeic species often occurring in wet/humid habitats which lays its eggs in the forest floor (*Van Heerdt, Blokhuis & Van Haaften, 1976*), the eggs need to be protected against drought or flood. Thus, it cannot be excluded that additional abdominal setae maybe used to estimate habitat conditions. A lack or a lower number of such setae in the case of males of *P. oblongopunctuatus* can be a good confirmation of this hypothesis as their role in reproductive behaviour ends with the act of copulation (*Thornhill & Alcock, 1983*). Although detailed studies upon the role of elytral setae of this ground beetle species should be done in the future, similar conclusions pointing to modified sexual behaviour based on atmospheric pressure changes (even if based on hygroreceptors located on antennae), which of course are connected to air humidity, were drawn for other insects (*Pellegrino et al., 2013*).

From the viewpoint of systematics, it is of interest that in the presented study individuals of *P. oblongopunctatus* had up to four pits in the fifth interval of the elytra. However, in 2010 the first author detected a female with even seven pits in one of the fifth intervals. The highest number of pits detected in the seventh row of elytra was two. Therefore, the possibility of numerous pits in the third interval of the elytra and pits in the seventh interval of elytra should be added to identification keys.

## CONCLUSIONS

The appearance of pits in the fifth and seventh interval of the elytra of *Pterostichus oblongopunctatus* seems to be related to moisture conditions. Since currently there is a lack of clear evidence that the setae in elytral pits in *P. oblongopunctatus* play a role of hygroreceptors, detailed studies upon the role of these setae should be done in the future. Moreover, the possibility of numerous pits in the third interval of the elytra and pits in the seventh interval of elytra should be added to identification keys.

## ACKNOWLEDGEMENTS

We thank Andrey Matalin and an anonymous reviewer for valuable comments on the manuscript. This paper is communication no. 494 of the Laboratory of Evaluation and Assessment of Natural Resources, Warsaw University of Life Sciences–SGGW.

### Funding
The studied material was elaborated within the scope of the project no. OR-2717-264 21/14, funded by General Directorate of the State Forests (DGLP). The funders had no role in study design, data collection and analysis, decision to publish, or preparation of the manuscript.

### Grant Disclosures
The following grant information was disclosed by the authors:
General Directorate of the State Forests (DGLP): OR-2717-264 21/14.

### Competing Interests
The authors declare there are no competing interests.

### Author Contributions
- Axel Schwerk conceived and designed the experiments, performed the experiments, analyzed the data, contributed reagents/materials/analysis tools, prepared figures and/or tables, authored or reviewed drafts of the paper, approved the final draft.
- Radomir Jaskuła analyzed the data, prepared figures and/or tables, authored or reviewed drafts of the paper, approved the final draft.

### Field Study Permissions
The following information was supplied relating to field study approvals (i.e., approving body and any reference numbers):

Field experiments were approved by the Skarb Państwa—Dyrekcja Generalna Lasów Państwowych. Approval number: OR-2717-21/14.

### Data Availability
The raw data is provided as Data S1.

### Supplemental Information
Supplemental information for this article can be found online at http://dx.doi.org/10.7717/peerj.4657#supplemental-information.

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
