# Peer review of "Rare patterns of dorsal puncture in Pterostichus oblongopunctatus (Coleoptera: Carabidae)"

_PeerJ, doi:10.7717/peerj.4657_

## Round 0.1 · original submission · Minor Revisions

Dear Radomir,

I thank you very much for you contribution that needs a few minor revisions prior to publication. Please see the appended Reviewer reports.

Reviewer 1 ·

Basic reporting

The English language in the article corresponds to all requirements to international publications. The style of the article is clear and concise.

The literature sources are appropriate and recent, include most publications on the studied species of ground beetles.

Structure conforms to PeerJ standards, corresponds to the standards of entomological experimental publications.

The figures are of good quality, contrasting, organized in accordance with the norms of international publications.

Experimental design

The original research is based on extensive collections of one species of ground beetles. The study is in the scope of our Journal.

The studied issue is relevant and is discussed in many articles, and is studied partly for the studied species of ground beetles; therefore the reviewed article is of great value.

The study corresponds to all ethic standards; the invertebrates were collected with minimal negative impact on the nature ecosystems. The selection is sufficient for formulating the presented conclusions. The method of collecting of the ground beetles adequately reflects the structure of the natural populations of P. oblongopunctatus. The study was financed by the General Directorate of the State Forests, which suggests that the field work was conducted in accordance with the required permissions.

The field and laboratory methods were appropriate, and the data on frequency are grouped in the Table. The frequency of the studies was sufficient.

Validity of the findings

Rational replication is accepted.

The data are analyzed correctly using Chi-square-tests and Mann-Whitney U-test. Data are robust. The data in the table are clear, detailed, and acknowledge the conclusions.

Conclusions are well stated and based on the actual results.

The text has no speculations. The hypotheses and theories in the discussion are substantiated, and the required links to the sources are provided.

The article is organized correctly and practically has no typing errors or inaccuracies. The study makes a good impression and can be recommended for publication.

·

Basic reporting

I think, the papers of V.M. Emetz, 1984a (Zoologicheskii Zhurnal, 1984, Vol.63, No.2, P.218-221), and 1984b (Ecologia, 1984, No.5, P.85-88) could be discussed within this study too.

Experimental design

No comment.

Validity of the findings

No comment.

Additional comments

I think lines 114-120 of the chapter “Statistical methods” can be deleted, because the numbers of damaged specimens are few. Thus, in the Table 1 the information about damaged specimens as well as about specimens with unclear sex can be deleted as well.

In lines 154-155 of the chapter “Results” indicated “Altogether, 1071 specimens (346 individuals in 1014, 417 in 1015, and 308 in 1016) of Pterostichus oblongopunctatus were analysed”. However, according to Table 1 the numbers of specimens is not the same: total – 1057; 2014 – 343; 2015 – 413; 2016 – 302. Why? Please check the numbers of the specimens.

In lines 215-217 of the chapter “Discussion” indicated “The result indicating that individuals with unusual puncture patterns had more pits in general, even in the third interval of the elytra, supports the hypothesis that those patterns are related to the moisture conditions during the period of larval development”. But in the MS the information about correlation of the numbers of elytral pits and the moisture conditions during the period of larval development is not presented.

In lines 224-227 of the chapter “Discussion” indicated “A higher number of abdominal setae due to a higher number of pits in the fifth and seventh intervals in females compared to males, as shown in our study, may be explained by the role of this sex in the reproductive behaviour”. However, the information about numbers of abdominal setae in the specimens with usual and unusual numbers of elytral pits as well as the correlation between numbers of abdominal setae and numbers of elytral pits is not presented in the chapter “Results”.

The MS is clearly written in good language.

---

## Round 0.2 · accepted · Accept

I think that the manuscript is now ready to be published.

#